# *Encephalitozoon cuniculi* takes advantage of efferocytosis to evade the immune response

**Luciane Costa Dalboni**[1], **Anuska Marcelino Alvares Saraiva**[2,3], **Fabiana Toshie de Camargo Konno**[1], **Elizabeth Cristina Perez**[1], **Jéssica Feliciana Codeceira**[1], **Diva Denelle Spadacci-Morena**[3], **Maria Anete Lallo**[1]*

**1** Programa de Patologia Ambiental e Experimental da Universidade Paulista–Unip, São Paulo, Brazil,
**2** Mestrado e Doutorado Interdisciplinar em Ciências da Saúde da Universidade Cruzeiro do Sul, São Paulo, Brazil, **3** Laboratório de Fisiopatologia, Instituto Butantan, São Paulo, Brazil

* anetelallo@hotmail.com, maria.lallo@docente.unip.br

**Data Availability Statement:** All relevant data are within the manuscript and its Supporting Information files.

## Abstract

Microsporidia are recognized as opportunistic pathogens in individuals with immunodeficiencies, especially related to T cells. Although the activity of CD8+ T lymphocytes is essential to eliminate these pathogens, earlier studies have shown significant participation of macrophages at the beginning of the infection. Macrophages and other innate immunity cells play a critical role in activating the acquired immunity. After programmed cell death, the cell fragments or apoptotic bodies are cleared by phagocytic cells, a phenomenon known as efferocytosis. This process has been recognized as a way of evading immunity by intracellular pathogens. The present study evaluated the impact of efferocytosis of apoptotic cells either infected or not on macrophages and subsequently challenged with *Encephalitozoon cuniculi* microsporidia. Macrophages were obtained from the bone marrow monocytes from C57BL mice, pre-incubated with apoptotic Jurkat cells (ACs), and were further challenged with *E. cuniculi* spores. The same procedures were performed using the previously infected Jurkat cells (IACs) and challenged with *E. cuniculi* spores before macrophage pre-incubation. The average number of spores internalized by macrophages in phagocytosis was counted. Macrophage expression of CD40, CD206, CD80, CD86, and MHCII, as well as the cytokines released in the culture supernatants, was measured by flow cytometry. The ultrastructural study was performed to analyze the multiplication types of pathogens. Macrophages pre-incubated with ACs and challenged with *E. cuniculi* showed a higher percentage of phagocytosis and an average number of internalized spores. Moreover, the presence of stages of multiplication of the pathogen inside the macrophages, particularly after efferocytosis of infected apoptotic bodies, was observed. In addition, pre-incubation with ACs or IACs and/or challenge with the pathogen decreased the viability of macrophages, reflected as high percentages of apoptosis. The marked expression of CD206 and the release of large amounts of IL-10 and IL-6 indicated the polarization of macrophages to an M2 profile, compatible with efferocytosis and favorable for pathogen development. We concluded that the pathogen favored efferocytosis and polarized the macrophages to an M2 profile, allowing the survival and multiplication of *E. cuniculi* inside the macrophages and explaining the possibility of macrophages acting as Trojan horses in microsporidiosis.

**Funding:** This paper received a grant from Fundação de Amparo à Pesquisa do Estado de São Paulo, grant number 2015/25948-2 (MAL). LCD received scholarships from Coordenação de Aperfeiçoamento de Pessoal de Nível Superior and Coordenação de Aperfeiçoamento de Pessoal de Nível Superior to JFC. The funders had no role in study design, data collection and analysis, decision to publish, or preparation of the manuscript.

**Competing interests:** The authors have declared that no competing interests exist.

## Introduction

The phylum Microsporidia comprises a diverse group of single-celled, spore-forming, obligatory intracellular pathogens that, since their identification for over 150 years, are widely distributed in nature as an etiological agent of pebrine, a disease occurring in silkworms [1]. More than 200 genera and 1,400 microsporidian species are already described and are phylogenetically related as a sister clade in the Fungi kingdom [2]. Microsporidia are well known as the most efficient and developed intracellular agents for parasitism of invertebrates and vertebrates. There exists a balanced relationship between host and pathogen, resulting in mild clinical diseases in immunocompetent individuals; however, severe disseminated infections can prove lethal for immunosuppressed individuals. Therefore, considering the advent of AIDS becoming a pandemic; human and animal healths are at risk due to these opportunistic agents [3]. Moreover, microsporidia affect immunosuppressed individuals due to neoplasms, chemotherapy, or anti-transplant treatment, including children and the elderly [4–7].

*Encephalitozoon cuniculi* are the first microsporidia that were successfully cultivated [8]. It causes infection in epithelial and endothelial cells and fibroblasts and macrophages of a wide variety of mammals. Moreover, it has been extensively used to understand various aspects of the pathogenesis of microsporidiosis [5]. Microsporidia enter the host cells by extruding the polar spore tubule or by phagocytosis/endocytosis of spores via macrophages and other phagocytes. Once inside the host cell, the pathogen develops until the new spores are formed, thereby causing cell breakdown [9]. Local macrophages, especially those in the digestive tract, rapidly recognize these microsporidia through various classes of receptors, including standard recognition receptors (PRRs). There is a possibility that these receptors are Toll-like receptor 2 (TLR2) and TLR4, which activate NF-κB (nuclear factor Kappa B) and increase the secretion of chemokines while recruiting more number of macrophages from monocytes. In addition, the recognition of microsporidia by macrophages could result in producing numerous other defense mediators, including cytokines and reactive oxygen and nitrogen species [10,11]. Microsporidia evade this protective response, whereas macrophages carrying these pathogens throughout the body and infecting new cells become true Trojan horses; however, the mechanisms involved in this type of response are still unknown and can be decisive for the course of the infection [12].

Furthermore, CD8+ T lymphocytes play an important role in defending against intracellular pathogens such as microsporidia [13,14]. After recognizing specific antigens, CD8+ T cells can use three mechanisms to kill the infected cells. First, the production of cytokines, especially IFN-γ and TNF-α, which interfere with the replication of microorganisms and antitumor activity, induces apoptosis. Second, the production and secretion of cytotoxic granules with granzymes and perforins where perforins form the pores in the membrane of target cells, followed by granzyme penetration, initiate apoptosis of infected cells via Caspase 8. Lastly, the mechanism of cell death involves the Fas/FasL pathway. The activated CD8+ T cell expressing FasL on its surface binds to the target cell's Fas receptor and undergoes a molecular modification that activates the cascade of caspases and apoptosis [15]. Previous studies report the death of infected cells mediated by perforins and granzymes in the KO mice when tested experimentally against the microsporidia infection by *E. cuniculi* [13]. Conversely, the CD8 mice−/−− treated with albendazole survived the entire experimental period despite having an enormous load of microsporidia. This confirms that the current treatments can be considered effective, where CD8+ T lymphocytes play a major role; however, these are not the only factor in the effective control of microsporidiosis [16].

Programmed cell death plays an important function in innate and acquired immunity to eliminate the infected cells and control the infection [17]. In contrast, certain pathogens block

or delay the death of host cells by promoting their intracellular replication at the beginning of the infection. This causes leakage and dissemination, eventually lysing the host cells. In certain cases, cell death pathways are co-opted by pathogens as a pathogenesis strategy. In addition, the pathogens can eliminate the primary defense cells by inducing the death of the host cells and, consequently, evade the host's defenses. Thus, the death of infected cells has a relevant role in the course of the infection and the immune response, with subsequent inflammation, resulting in efferocytosis [18].

Initially, efferocytosis was defined as the phagocytosis of apoptotic bodies that may or may not be infected by phagocytes [19]. Recently, this concept has expanded to involve phagocytosis of other cells killed by programmed necrosis that has a crucial role in the late events of inflammation. Furthermore, efferocytosis suppresses the immune response by releasing anti-inflammatory mediators such as IL-10, TGF-β, prostaglandin E2 (PGE2), as well inhibits the synthesis of pro-inflammatory mediators such as TNF-α, GM-CSF, IL-12, IL-1β, IL-18, and LTC4 [20,21]. Phagocytosis of apoptotic polymorphonuclear cells with the internalized pathogens can either kill the pathogen or allow the transfer of the ingested pathogen via neutrophils, acting as a transport vehicle, to the macrophages, as observed previously in intracellular pathogens, such as *Leishmania major* [22], *Chlamydia pneumoniae* [23], and *Burkholderia pseudomallei* [24].

With these specifications, efferocytosis has been investigated as a mechanism for evading the immune response by infectious agents. However, for *M. tuberculosis* (Mtb), the efferocytosis of apoptotic and infected macrophages increased the microbicidal activity of macrophages [18,25,26], whereas Mtb was released by macrophages that underwent programmed necrosis and infected and proliferated in the other macrophages, enabling the spread of the disease [27], with necrosis being the type of death that favors this pathogen [28]. We demonstrated that efferocytosis of infected apoptotic cells had no suppressive effect on the macrophage microbicidal activity with the multiplication of *E. cuniculi* inside the macrophages. This converted the macrophages to an M2 profile with the production of large amounts of IL-10 and IL-6, corroborating the hypothesis that this pathogen could be exploited in the less inflamed environment present in efferocytosis to evade an immune response.

## Materials and methods

### Ethics approval

The Ethics Committee of Animal Care and Experimentation of Universidade Paulista-Unip approved this study (approval number 010/17). Mice were treated in accordance with the Guiding Principle for Animal Care and Experimentation of CONCEA.

### *E. cuniculi* spores

Spores of *E. cuniculi* (genotype I) (Waterborne Inc., New Orleans, LA, USA) used in this experiment were previously cultivated in a rabbit kidney cell lineage (RK-13, ATCC CCL-37) in Eagle's medium supplemented with 10% of fetal calf serum (FCS) (Cultilab, Campinas, SP, Brazil), pyruvate, nonessential amino acids, and gentamicin at 37°C in 5% $CO_2$. The spores were purified by centrifugation, and cellular debris was excluded by 50% of Percoll (Pharmacia) as described previously [6]. The proportion of two spores per cell (2:1) were used for infection.

### Marrow donor mice

Specific pathogen-free (SPF) inbred C57BL/6 female mice with 6 to 8 weeks of age were obtained from the animal facilities of Centro de Desenvolvimento de Modelos Experimentais

(CEDEME), UNIFESP, Brazil. The animals were housed in polypropylene micro isolator cages with a 12/12 h light/dark cycle, maintained at 21 ±2˚C and >40% humidity, and had standard chow and water *ad libitum*. All experimental procedures were performed in accordance with the animal care guidelines.

## Macrophage isolation from bone marrow precursors

The animals were euthanized with anesthetic deepening using ketamine (100 mg/mL), xylazine (20 mg/mL), and fentanyl (0.05 mg/mL). The femurs and tibiae of mice were collected aseptically, and the bone marrow was extracted by infusing 10 mL of phosphate-buffered saline (PBS). The cell aggregates were disrupted carefully by homogenization. The cell debris was subsequently eliminated using hemolytic buffer ammonium chloride/Tris Buffer (ACT), with a concentration of 160 mM $NH_4CL$ and 170 mM TRIS-base. After washing with RPMI, the cell suspensions were grown in R10 buffer (RPMI + 10% SFB + 1% gentamicin) with the addition of L929 cell supernatant to stimulate the macrophage differentiation. The cultures were maintained at 37˚C in an atmosphere of 5% $CO_2$ for 5 to 7 days, with an additional 8 mL of this medium added on the fourth day of incubation. On the seventh day, the macrophages were put in an ice bath for 15 min and subsequently dropped in RPMI using a cell scraper. Next, the macrophages were cultured at a concentration of $2 \times 10^5$ cells per well in 24-well plates. The cell viability was checked by Trypan blue, which was at least 90%.

## Viability and phenotype of macrophages isolated from bone marrow

Cell viability was analyzed after cultivation with Kit 7AAD/Annexin (BD Pharmingen), according to the manufacturer's instructions. Briefly, the cells were centrifuged and resuspended in the Kit buffer, followed by the addition of 1 μL of annexin and 1 μL of 7AAD. After 15 min of incubation at room temperature in the dark, the kit buffer was again added, and apoptosis was quantified on an Accuri$^{TM}$C6 flow cytometer (BD Biosciences, Mountain View, CA) while collecting from each tube after 30 s. For phenotyping of macrophages in profile M1 or M2, the non-specific Fc receptors were blocked with anti-CD16/32 antibodies and incubated for 15 min. After washing with Mac's buffer (0.5% BSA and 0.075% EDTA), the macrophages were labeled with mouse anti-F4/80 monoclonal antibody conjugated to phycoerythrin (PE), mouse anti-MHC II monoclonal antibody conjugated to fluorescein isothiocyanate (FITC), mouse anti-CD80 and anti-CD86 monoclonal antibodies conjugated to FITC, mouse anti-CD40 monoclonal antibody conjugated to PECy5, and mouse anti-CD206 monoclonal antibody conjugated to Alexa Fluor 647 for 30 min at 4˚C. Once again, cells were washed with Mac's buffer resuspended in 200 μL of PBS to assess their viability and phenotype them on an Accuri$^{TM}$C6 flow cytometer with 200,000 events collected for each sample for 30 s.

## Jurkat cell culture

Jurkat cells from the ATCC strain were kindly donated by Prof. Alexandra Ivo de Medeiros (Department of Biological Sciences, Basic Immunology laboratory at UNESP in Araraquara) and stored at–80˚C in a freezer. For each experiment, the cells were thawed, centrifuged for 5 min at 1,500 rpm, homogenized, and resuspended in 10 mL of R10 medium with 5% $CO_2$ at 37˚C. Ten Jurkat cells were used for each macrophage (10:1) in the efferocytosis assays.

## Ultraviolet (UV) radiation-induced apoptosis from Jurkat cells

Jurkat cells were centrifuged for 5 min at 1,500 rpm and resuspended in 5 mL of R10, homogenized, and counted to obtain a sufficient number for the experiment. To induce apoptosis, 100

J of UV radiations were optimized using the UVC-500 Ultraviolet Crosslinker (Amersham, Biosciences USA). After 4 h of incubation, apoptosis was quantified with the 7AAD/Annexin Kit (BD Pharmingen) according to the manufacturer's instructions as previously described. For this study, 100 J of UV light was selected, with an average of 40% apoptosis and about 15% of dead cells (S1 Fig).

## Jurkat cell infection and apoptosis

Jurkat cells were infected (IACs) with *E. cuniculi* spores in the proportion of two spores per cell. After 90 min of incubation, the cells were centrifuged at 1,500 rpm for 5 min; the pellet was resuspended in the medium and subjected to 100 J of radiations to induce apoptosis. For all experiments, the percentage of apoptosis was verified using the 7AAD/Annexin kit, as previously described.

## Experimental design

Macrophages were pre-incubated with apoptotic Jurkat cells that were either previously infected with *E. cuniculi* spores or not, for 1 h, for efferocytosis to occur. Next, the supernatant was discarded and washed with PBS, followed by challenging the cultures with *E. cuniculi*, with two spores per macrophage. After 1 h, the supernatant was discarded, and the wells were washed with PBS, then 1 mL of R10 per well was added, maintained for 1 h, 12 h, and 24 h to observe the events resulting from efferocytosis (Fig 1).

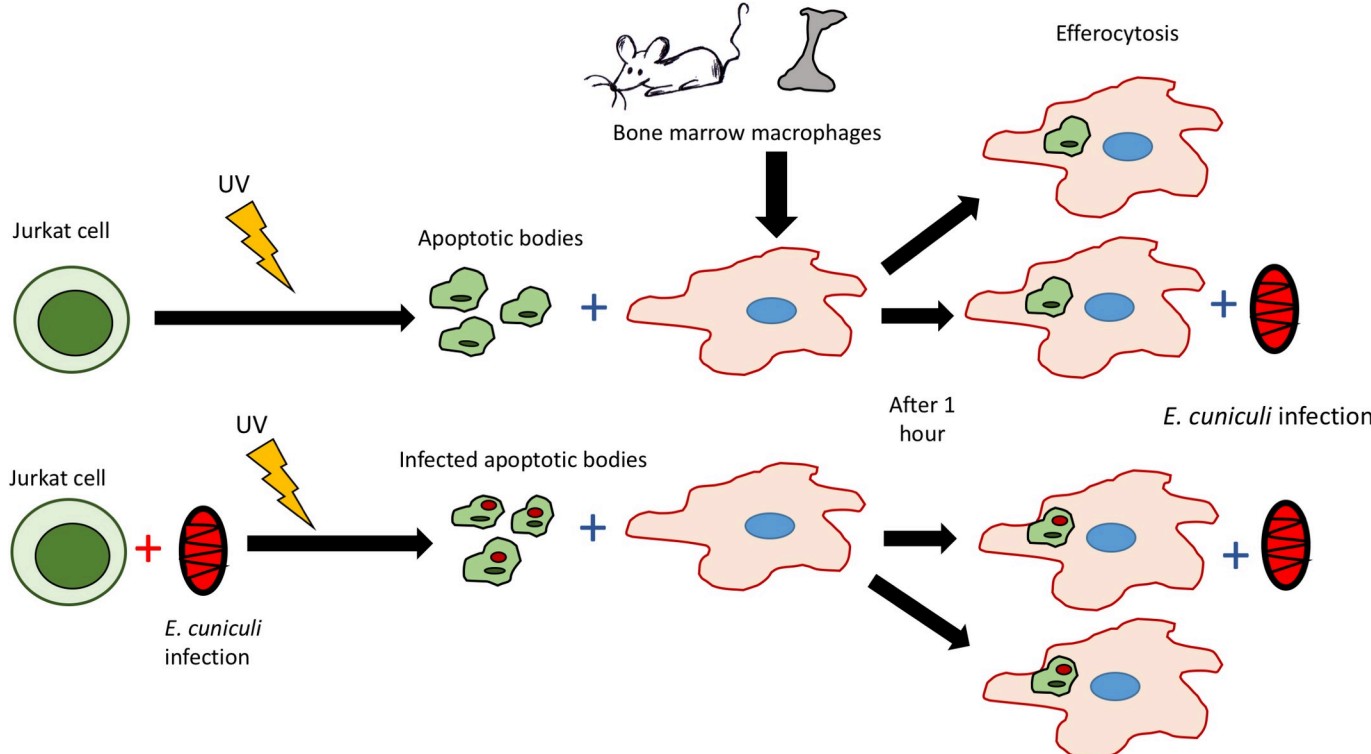

**Fig 1. Experimental design showing that macrophages were pre-incubated with apoptotic Jurkat cells, previously infected or not with *E. cuniculi*.** After efferocytosis, they were challenged with infection by *E. cuniculi* in a total of four groups: Macrophages with *E. cuniculi*; macrophages with ACs; macrophages with ACs and *E. cuniculi*; macrophages with IACs and *E. cuniculi*.

## Quantification of phagocytosis of *E. cuniculi* spores by macrophages

After 1 and 24 h of incubation, the coverslips were stained with Calcofluor, and the spores were internalized and counted in 100 macrophages under a fluorescence microscope (Olympus-BX60 Tokyo, Japan). The percentage of macrophages that phagocytosed *E. cuniculi* spores and the average number of internalized spores per macrophage were obtained and statistically analyzed.

## Determination of cytokine profile

The release of cytokines by macrophages pre-incubated with ACs was studied using the supernatant of cultures challenged or not with *E. cuniculi* at 1 h, 12 h, and 24 h. An inflammation kit (BD Biosciences, CA, USA) was used to detect the levels of MCP-1, IL-12, p70, IL-6, IL-10, IFN-γ, and TNF-α. Briefly, 25 μL of each sample was incubated with capture beads conjugated to APC and with the detection antibody conjugated to PE for 2 h at room temperature in the dark. Subsequently, the samples were washed with a wash buffer, centrifuged, and resuspended in the same buffer for two-color analysis using the BD Accuri™ C6 flow cytometer (BD Biosciences, Mountain View, CA). The analysis was performed using the FCAP array software v3.0.

## Quantification of phagocytosis of apoptotic Jurkat cells by macrophages

To quantify the phagocytosis of Jurkat cells infected or not by macrophages, the Jurkat cells were labeled with carboxyfluorescein diacetate succinimidyl ester (CFSE, BD-horizon). Briefly, 1 μL of CFSE was diluted in 5 mL of 5 mM PBS, and then 1 mL of this solution was added up to $30 \times 10^6$ Jurkat cells for 10 min at 37°C. Subsequently, the cells were washed with 10 mL of PBS and resuspended in R10, and irradiated with an energy pulse of 100 J of UV light. After 4 h of incubation, the cells were centrifuged and cultured with macrophages for 1 h. The macrophages were unleashed, as previously described, in obtaining the macrophages from bone marrow precursors. After blocking the non-specific Fc receptors, they were marked with mouse anti-F4/80 monoclonal antibodies conjugated to allophycocyanin (APC) for reading in an Accuri™ C6 flow cytometer, collecting 200,000 events for 30 s in each sample. Apoptotic cells that were not phagocytized were excluded, and the data obtained were analyzed using the PRISMA program.

## Viability and phenotypes of macrophages pre-incubated with Jurkat cells and infected with *E. cuniculi*

Viability and phenotypes of macrophages were studied at 1 h and 24 h. Macrophages pre-incubated or not with ACs (infected Jurkat or uninfected Jurkat) were incubated in the presence or absence of *E. cuniculi*. The viability was measured using the 7 AAD/Annexin kit. The macrophages were phenotyped for profiles M1 and M2, as discussed previously. The reading was performed on an Accuri™ C6 flow cytometer (BD Biosciences, Mountain View, CA), collecting 200,000 events for 30 s in each sample.

## Transmission electron microscopy

Differentiated macrophages were grown in 25 cm² bottles and divided into the following groups: macrophage infected with *E. cuniculi*; macrophages pre-incubated with ACs; macrophages pre-incubated with ACs and infected with *E. cuniculi*, and macrophages pre-incubated with IACs and infected with *E. cuniculi*. After 24 h, the cultures were collected and fixed in 2% glutaraldehyde in 0.2 M cacodylate buffer (pH 7.2) at 4°C for 10 h, and subsequently post-

fixed in 1% OsO$_4$ buffered for 2 h. Semi-thin and ultrathin sections stained with toluidine blue were viewed under a light microscope and TEM, respectively, and were photographed using the LEO EM 906E electron microscope at 80 kV.

### Analysis of internalization of spores by Jurkat cells

Jurkat cells were previously cultured and marked with a fluorescent dye, PKH-26 (Sigma Aldrich, St Luis, MO, USA), according to the manufacturer's instructions. The cells were centrifuged for 5 min at 2,000 rpm, following which the supernatant was discarded. To the pellet, 2 μL of the PKH 26 dye was added in 500 μL of the C diluent (PKH kit) and incubated for 5 min at room temperature. Following this, the pellet was washed thrice with RPM. The *E. cuniculi* spores were marked with 0.1 μL of CFSE in 1 mL of 10 mM PBS and incubated for 37 min in the dark. After washing with PBS, the spores were inoculated into the bottles containing the PKH-labeled Jurkat cells, in a proportion of two spores per cell. The infection time was 1 h and 30 min followed by centrifugation. The supernatant and the cell pellet were subsequently placed on silanized slides and fixed with 4% PFA to preserve the cells, after which DAPI (Fluoroshield Sigma St Luis, MO, USA) was used to stain the core. The images were acquired and recorded using a Leica TCS SP8 confocal microscope and analyzed with LAS X software.

### Statistical analysis

Statistical analysis was performed using the GraphPad Prism 7 software, and comparison between the groups was performed using one-way or two-way analysis of variance (ANOVA). The values are presented as the average of the experimental replicates ±standard error with a 95% confidence interval.

## Results

### Multiplication of *E. cuniculi* in macrophages pre-incubated with apoptotic cells

The macrophages were incubated with apoptotic cells (ACs). These were next challenged with *E. cuniculi* spores to assess whether efferocytosis modified macrophage activity in microsporidiosis. After an hour of the challenge, about 50% of the macrophages internalized about two spores. Further, within 24 h, 80% of macrophages were observed in the phagocytosis with an average of two spores inside the macrophages (Fig 2), suggesting multiplication of the pathogen or macrophage death.

Apoptosis of infected cells is common in diseases caused by intracellular pathogens. The immune response involves the cytotoxic activity of CD8$^+$ T lymphocytes that cause phagocytosis of infected apoptotic bodies. Here, we assessed whether efferocytosis of infected Jurkat cells modulated the macrophage activity. The presence of *E. cuniculi* spores in the cytoplasm of Jurkat cells denoted infection, indicating that the pathogen was internalized by these lymphocytes (Fig 3). The percentage of apoptosis of infected Jurkat cells was similar to that in the uninfected group, indicating that the infection for a limited time did not change the susceptibility of these lymphocytes to apoptosis. Transmission electron microscopy after 24 h showed intact spores inside the macrophages in the phagosomes and amorphous material in the phagosomal vacuoles, suggesting spore lysis (Fig 3) and the formation of megasomes (S2 Fig). No form of pathogen development was recognized; therefore, multiplication of the pathogen within these macrophages could not be confirmed.

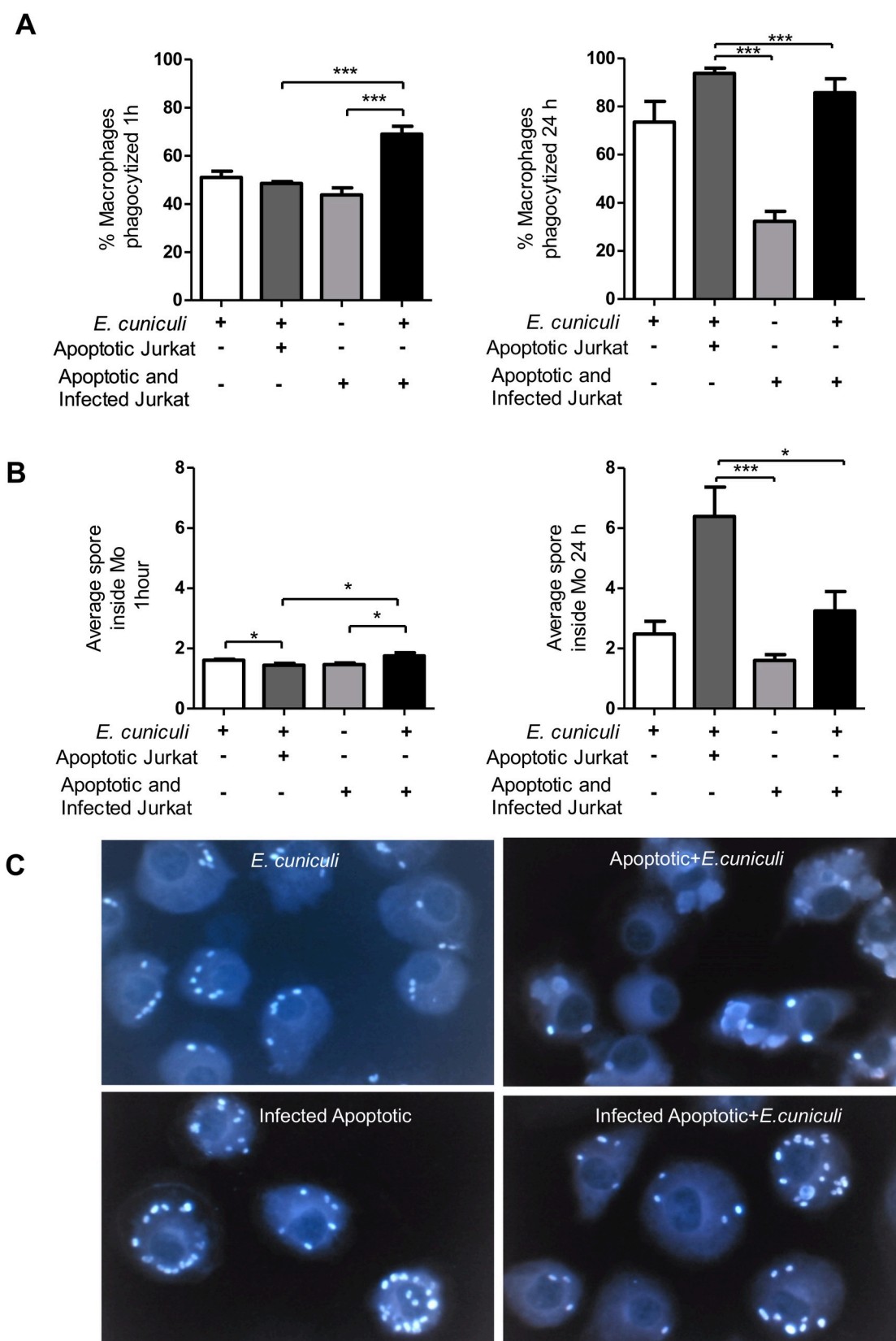

**Fig 2. Evaluation of the phagocytic activity of pre-incubated (+) or not (–) macrophages with apoptotic cells (ACs) or infected apoptotic cells (IACs) and challenged with *E. cuniculi*.** (A) Percentage of macrophages phagocyting the spores during 1 h and 24 h of incubation. (B) Average spores inside the macrophages in 1 h and 24 h of incubation. (C) Image of macrophages with spores of *E. cuniculi* stained by Calcofluor after pre-incubation with ACs or IACs. One-way analysis of variance (ANOVA) with Tukey's post-test revealed significance between the groups. $^*p < 0.05$, $^{***}p < 0.001$.

Macrophages were pre-incubated with ACs to determine the phagocytosis of apoptotic bodies, demonstrating the occurrence of efferocytosis (S3 and S4 Figs). In addition, necrosis cells were observed and characterized by the rupture of the cell membrane (S3 Fig).

When macrophages pre-incubated with ACs were challenged with *E. cuniculi*, the intact spore clusters were observed in the areas, suggestive of ruptured parasitophorous vacuoles (Fig 4A and 4B). Moreover, the spore was identified as starting the extrusion of the polar tubule (Fig 4C). These findings suggested a multiplication of the pathogen inside the macrophages, suggesting a higher percentage of phagocytosis and the average number of spores in the macrophages compared to the other groups evaluated in 24 h (Fig 2).

In addition, pre-incubation with ACs and/or the challenge with the pathogen decreased the cell viability of macrophages within an hour of onset of the infection (Fig 5A). The predominant pattern of death was apoptosis, with percentages ranging from 48 to 50% in 24 h when compared with the control (Fig 5B). However, when macrophages were pre-incubated with IACs, their viability decreased to about 50% within an hour of the challenge, with apoptosis being the most common type of death observed (Fig 5B).

Next, the macrophages were pre-incubated with infected apoptotic Jurkat cells (IACs). The ultrastructural analysis demonstrated the presence of apoptotic bodies containing the intact *E. cuniculi* spores in the macrophage cytoplasm (Fig 6). In this group, the spores were also observed next to a macrophage fragment during necrosis (Fig 6B) and macrophage apoptosis (Fig 6C). After 1 h, this group also had about 50% of the macrophages with an average of two internalized spores that were probably phagocyted together with the apoptotic bodies as they were not challenged (Fig 2). After 24 h, both phagocytosis and the average number of spores inside the macrophages decreased in relation to other groups pre-incubated with ACs (Fig 2).

In order to assess the influence of efferocytosis of the infected apoptotic cells or bodies, the macrophages were pre-incubated with IACs and subsequently challenged with *E. cuniculi* spores. In this group, we observed the presence of parasitophorous vacuoles containing forms of multiplication of *E. cuniculi*, such as immature merontes and spores, in addition to the clusters of mature spores, showing the multiplication of the pathogen in the macrophages (Fig 7).

About 80% of the macrophages underwent spore phagocytosis after 1 h and kept increasing until 24 h. The mean number of internalized spores (two spores per macrophage) was similar to the other groups that were challenged after 1 h. After 24 h, it increased to three spores per macrophage, with only below the average of the group that was exposed to ACs and was challenged (Fig 2).

## Profile of M2 macrophages with production of IL-10 associated with efferocytosis and challenged with *E. cuniculi*

One hour after the challenge, no change was observed in the fluorescence median (MFI) values for CD40 and CD206 molecules in all macrophage groups that were pre-incubated with ACs or not, as well as in the expression of the MHCII, CD80, and CD86 molecules (Fig 8A and 8B). However, 24 h after the challenge with *E. cuniculi*, the macrophages of infected groups showed a higher MFI for CD40 and CD206 in relation to their respective uninfected controls. In turn, macrophages challenged with *E. cuniculi* and pre-incubated with ACs had a higher expression

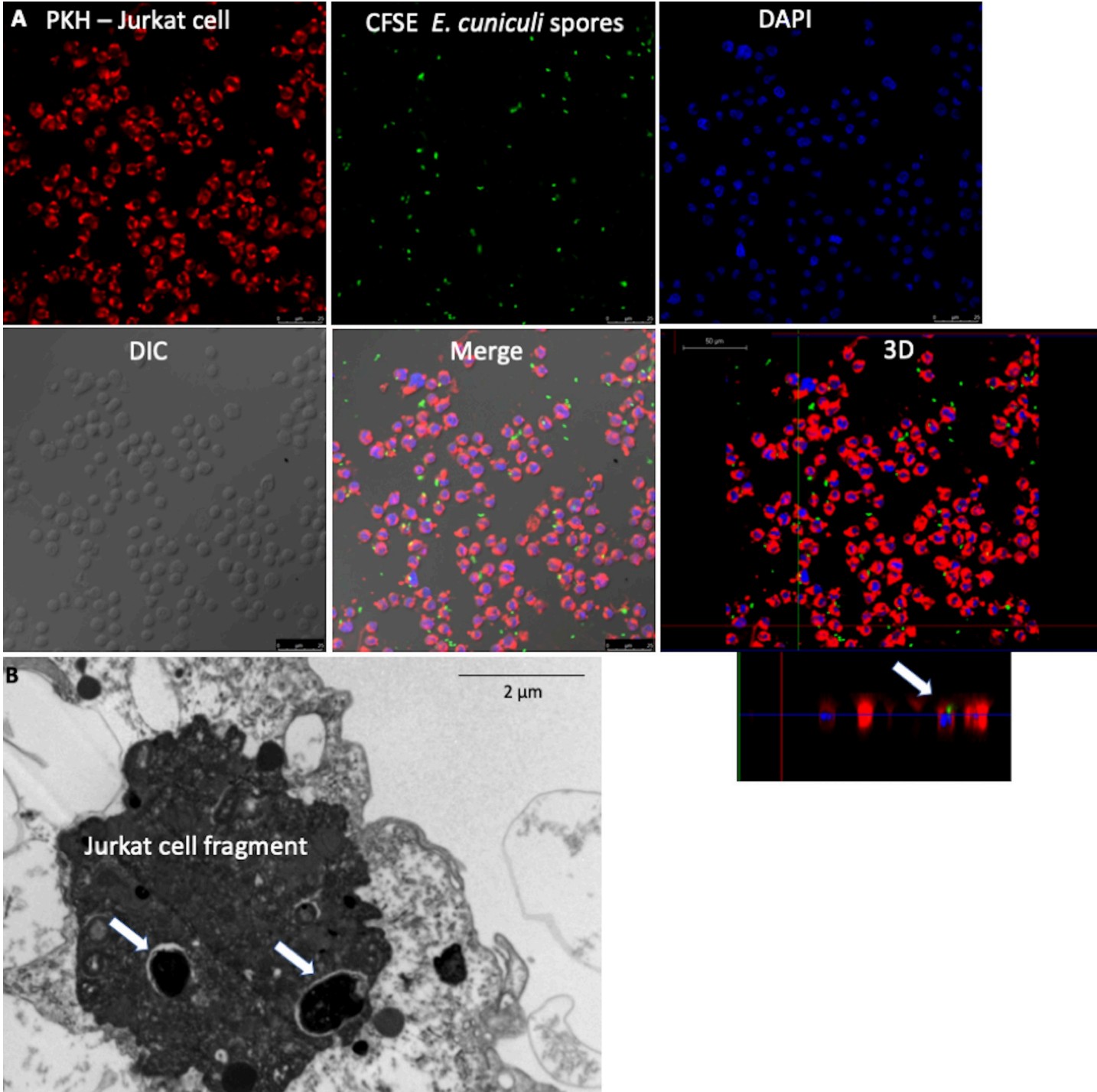

**Fig 3. Infection of Jurkat cells (ACs) by *E. cuniculi* before inducing apoptosis.** (A) Jurkat cells (ACs) stained with PKH; *E. cuniculi* spores marked with CFSE; core fluorescence stained with DAPI; phase-contrast cultures-DIC; overlay of images showing internalized and intact spores inside the Jurkat cells, confirmed in the 3D image (arrow). (Images) (B) Ultramicrographic images of a Jurkat cell fragment infected with *E. cuniculi* (arrow).

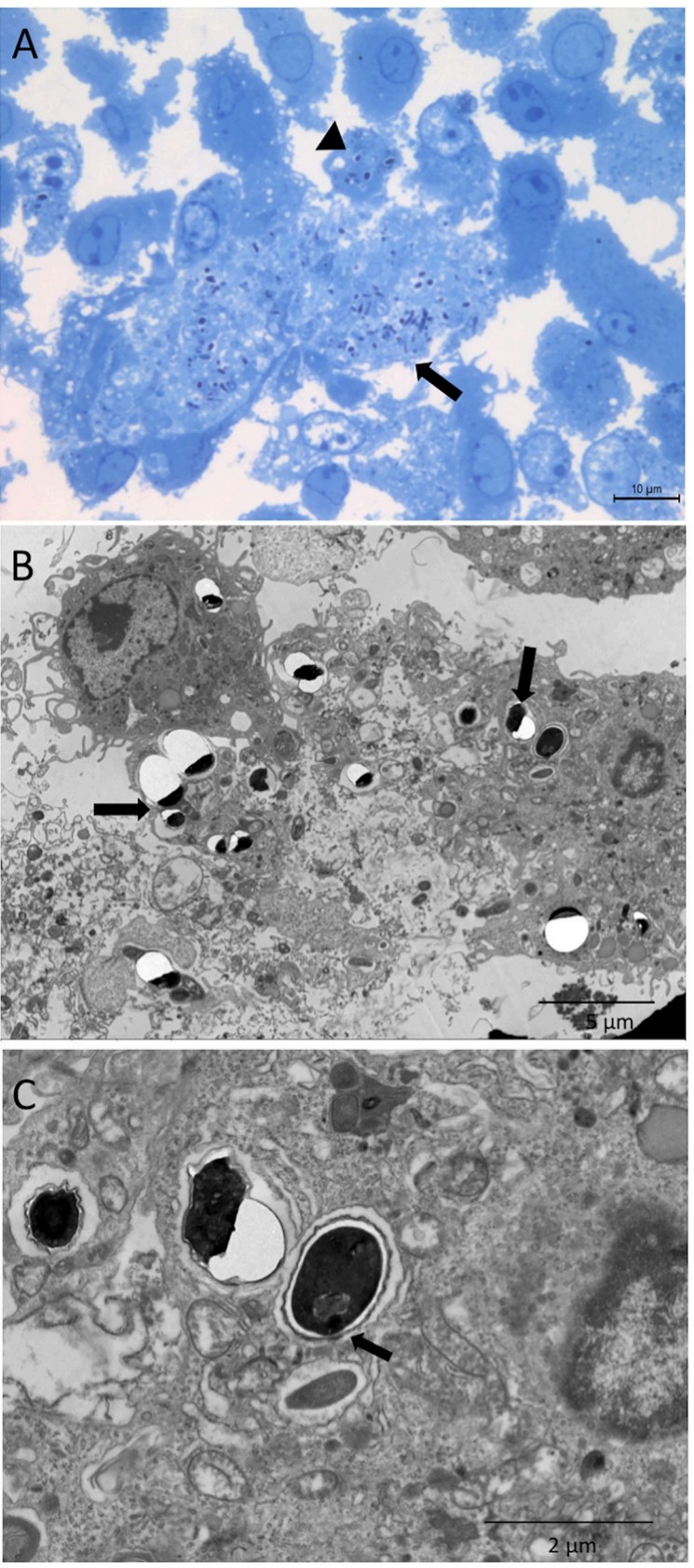

**Fig 4. Macrophages pre-incubated with apoptotic cells and challenged with *E. cuniculi*.** (A) Image showing clusters of *E. cuniculi* spores in a lighter area of the cytoplasm, suggesting a parasitophorous vacuole (arrows). Spores inside the macrophage with preserved characteristics (arrowhead) stained with toluidine blue. (B) Ultramicrography showing clusters of *E. cuniculi* spores in the macrophage cytoplasm with ruptured cell membrane following the multiplication of the pathogen (arrow). (C) Ultramicrography of *E. cuniculi* spores in the macrophage cytoplasm showing signs of polar tubule extrusion (arrow), as a sign of viability. Note that the rupture of the resin next to the spores is a characteristic finding of microsporidia.

of CD40, CD206, MHCII, CD80, and CD86 molecules as compared to the non-pre-incubated group.

When evaluating the ratio obtained between the MFIs of CD40^high^ and CD206^high^, a high fluorescence of CD40^high^ was observed through infection, indicating an M1 profile, when compared to the control. However, macrophages pre-incubated with non-challenged ACs showed a higher ratio as compared with the control group, suggesting polarization for the M1 profile.

In line with these findings, the infection by *E. cuniculi* determined the release of mixed cytokines, TNF-α, MCP-1, IL-6, and IL-10 when compared to the group of ACs not challenged with *E. cuniculi*, which exhibited cytokine-mediated inflammatory reactions without the release of the cytokine IL-10 and presenting an M1 profile. This release of mixed cytokines was enhanced by efferocytosis, as was observed in the groups pre-incubated with ACs and challenged, especially in the later periods with 12 and 24 h incubations. Moreover, the exclusive presence of the fungus, as well as the challenge-associated efferocytosis led to a greater release of IL-10, an anti-inflammatory cytokine (Fig 9).

Pre-incubation with IACs determined a higher expression of CD40 molecules on the surface of macrophages in one hour. After 24 h, all macrophages that came in contact with the pathogen, by challenge or by phagocytosis IACs, showed an increase in MFI for CD40, CD206, MHCII, CD80, CD86 in comparison with macrophages not challenged with *E. cuniculi*. However, a tendency to modulate these macrophages to an M2 profile was observed in pre-incubation with IACs. However, the mixed profile was evident of macrophages as

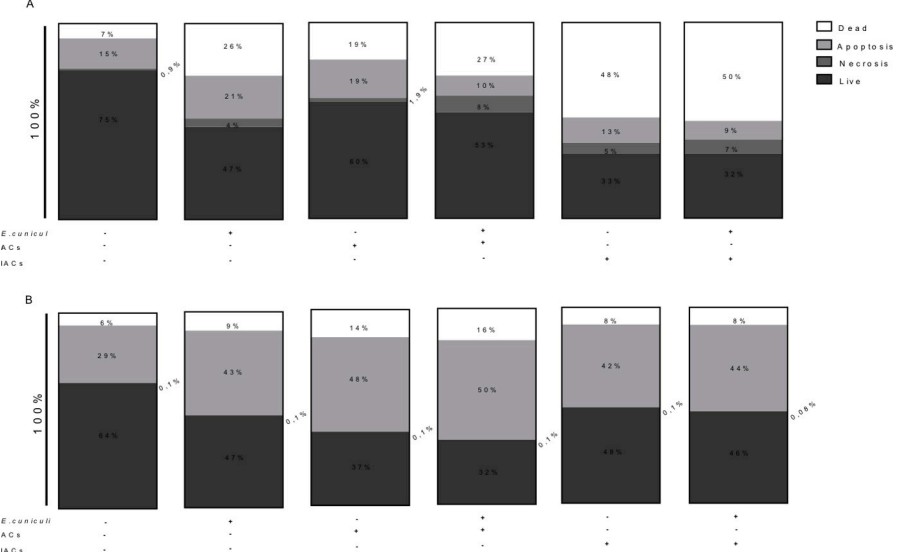

**Fig 5. Analysis of the viability of pre-incubated (+) or not (−) macrophages with apoptotic cells (ACs) or infected apoptotic cells (IACs) and challenged (+) or not (−) with *E. cuniculi*, after 1 h or 24 h of observation.** Percentages of dead, apoptotic, necrotic, and live cells (A) after 1 h of infection with *E. cuniculi* and (B) after 24 h of infection with *E. cuniculi*.

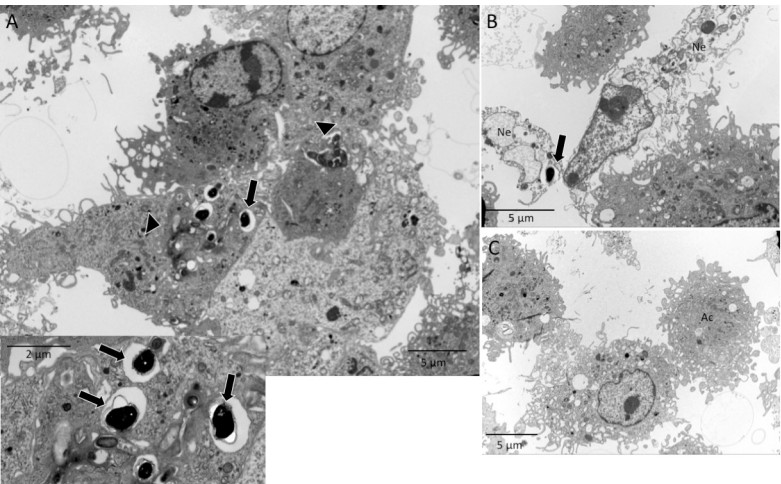

**Fig 6. Ultramicrography image of macrophages pre-incubated with *E. cuniculi* in infected and apoptotic cells.** (A) Apoptotic bodies phagocyted (arrowhead) by macrophages containing intact *E. cuniculi* spores inside (insert) (B) necrotic cells (Ne) with the spores of pathogen close to macrophages. (C) A macrophage in apoptosis (Ac).

compared to the other groups ([Fig 9]). The MFI of CD40$^{high}$ and CD206$^{high}$ in the macrophages was lower when compared to the other groups and was similar in comparison with the control group ([Fig 8]). These data suggested the double profiles of M1 and M2.

Macrophages pre-incubated with IACs increased the release of cytokines, TNF-α, MCP-1, IL-6, and IL-10, which intensified with time; this was similar to what was observed with macrophages challenged by *E. cuniculi*. An increase in the release of IL-10 was observed after 12 h in the group pre-incubated with IACs, whereas a significantly greater release was observed in

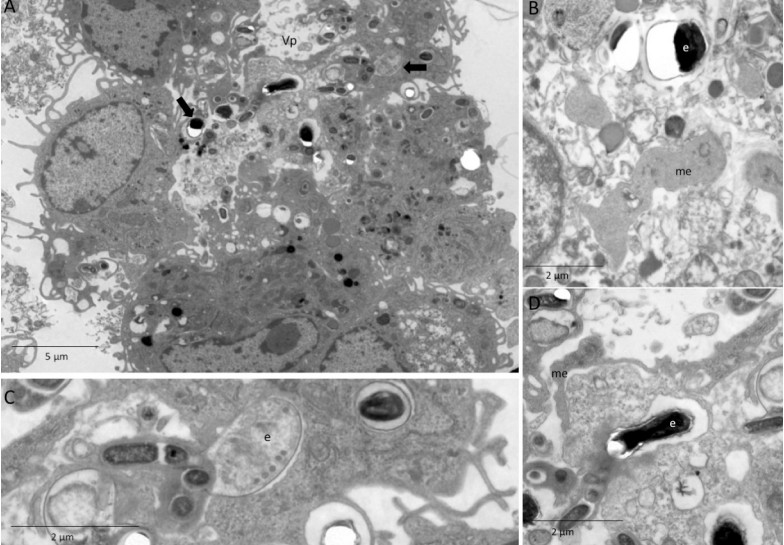

**Fig 7. Ultramicrography image of pre-incubation of macrophages with infected and apoptotic Jurkat cells after challenge with *E. cuniculi*.** (A) Parasitophore vacuole (Vp) with *E. cuniculi* spores (arrows) and other forms of development. (B) Detail of meronte (me) and spores (e) with rupture of resin, characteristic of microsporidia. (C) Detail of immature spores (e) showing an enameled polar tubule. (D) Detail of spores (e) with rupture of resin, characteristic of microsporidia and meronte (me) next to the Vp membrane.

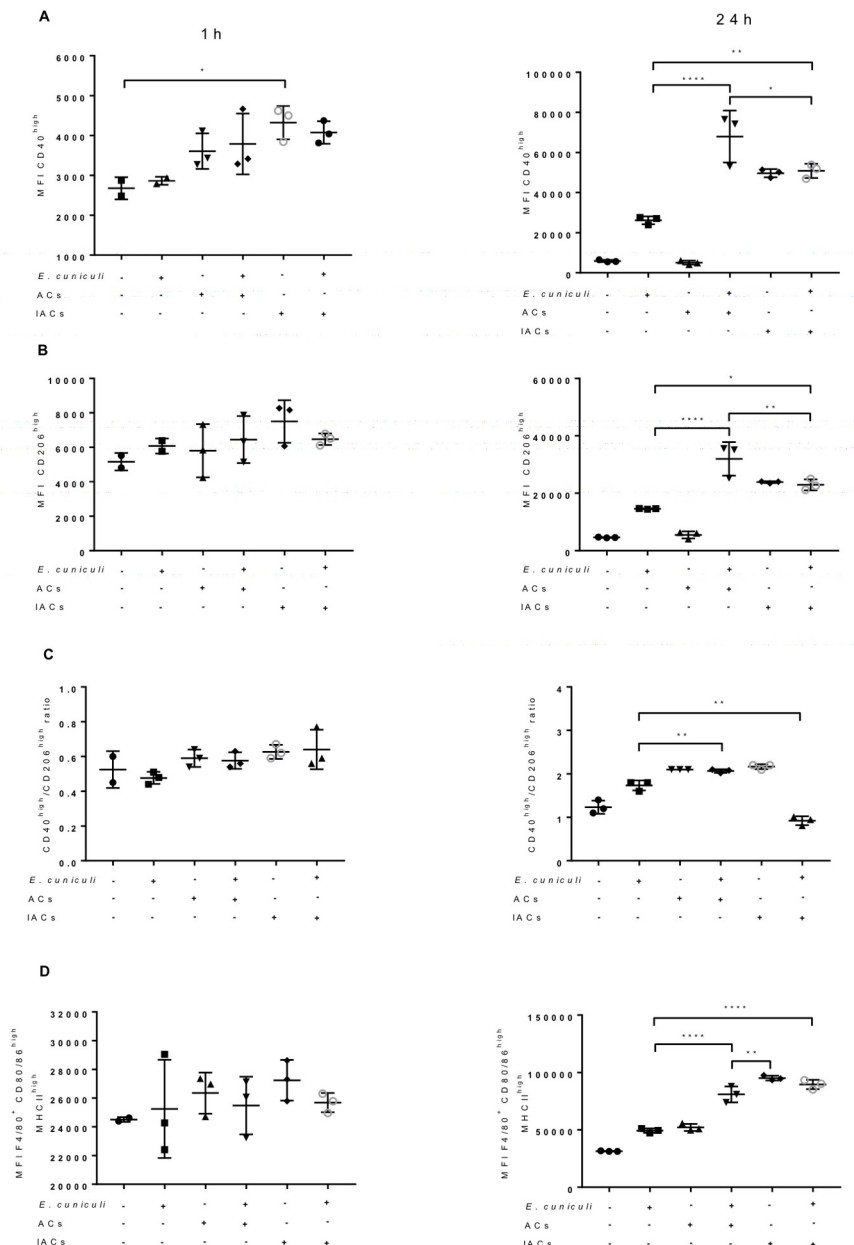

**Fig 8. Expression of polarization and activation surface markers in pre-incubated (+) or not (–) macrophages with apoptotic cells (ACs) or infected apoptotic cells (IACs) and challenged (+) or not (–) with *E. cuniculi*, after an hour or 24 h of observation.** (A) Median fluorescence (MFI) for CD40 in macrophages. (B) MFI for CD206 in macrophages. (C) The ratio between MFI CD40/MFI CD206. (D) MFI for CD80/86 and MHC II in macrophages. One-way analysis of variance (ANOVA) with Tukey's post-test revealed significance between the groups. $^*p < 0.05$, $^{**}p < 0.01$, $^{***}p < 0.001$.

the group with efferocytosis and challenged with the pathogen (Fig 9). The release of mixed cytokines corroborated with the double macrophage profile mentioned above.

## Discussion

Cell death following the infection by intracellular pathogens is a developed mechanism to reduce or prevent the replication and the spread of pathogens [29]. For the host, pathogen-

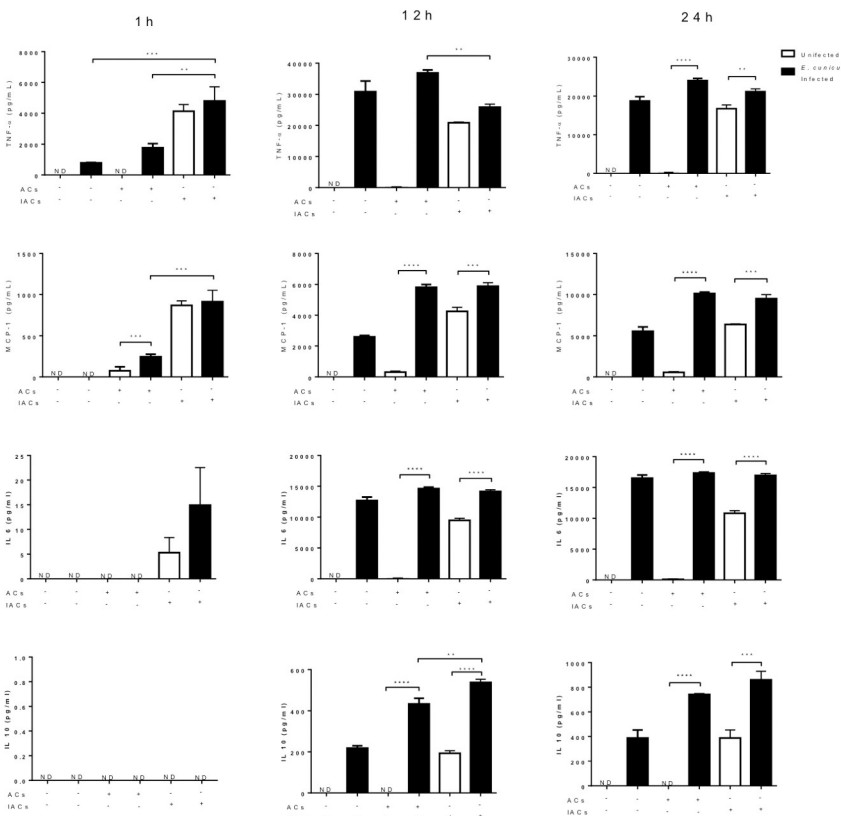

**Fig 9. Quantification of inflammatory (TNF-α, MCP-1, and IL-6) and anti-inflammatory (IL-10) cytokines released pre-incubated (+) or not (–) macrophages with apoptotic cells (ACs) or infected apoptotic cells (IACs) and challenged (infected *E. cuniculi*) or not (infected) in one hour, 12 and 24 h of observation.** One-way analysis of variance (ANOVA) with Tukey's post-test revealed significance between the groups. $^{*}p < 0.05$, $^{**}p < 0.01$, $^{***}p < 0.001$.

triggered death can (a) remove the intracellular environment necessary for the survival and replication of the pathogen; (b) direct microbicidal effects of the released intracellular components; (c) initiate an inflammatory antimicrobial response by presenting Damage Associated Molecular Patterns (DAMPs) and Pathogen-Associated Molecular Patterns (PAMPs), and (d) uptake and present pathogenic antigens by antigen-presenting cells (APCs) [18]. Secondary to cell death, efferocytosis involves phagocytosis of cells undergoing apoptosis, unscheduled necrosis, necroptosis, and pyroptosis. Subsequently, it promotes a favorable environment to prevent inflammation in several ways, such as downregulation of the expression of pro-inflammatory cytokines, inhibition of induced nitric oxide synthase, and increased levels of angiogenic growth factors, all of which are crucial for the survival of pathogens. In several cases, efferocytosis eliminates pathogens and infected cells. However, certain pathogens have subverted this process and use this mechanism to prevent innate immune detection and assist phagocyte infection; thus, acting as "Trojan horses" [18,30].

Herein, we observed that the macrophages pre-incubated with ACs or IACs underwent more phagocytosis and internalized a greater number of *E. cuniculi* spores. Moreover, it demonstrated various ways of developing the pathogen in their cytoplasms within parasitophorous vacuoles, allowing for the survival and replication of *E. cuniculi*. Corroborating our results, alveolar macrophages pre-incubated with apoptotic Jurkat cells and, subsequently, challenged

with *Streptococcus pneumonia* [31] showed multiplication of the pathogen. Similarly, macrophages pre-incubated with apoptotic *Leishmania major*-infected neutrophils enabled the protozoan to survive, demonstrating the low parasiticidal activity in the presence of ACs [22]. In contrast, macrophages that underwent the apoptotic Mtb-infected macrophages efferocytosis had total elimination of the pathogen [18,26] and showed decreased pathogen viability and increased antigenic presentation for an adaptive immune response [32]. Alternatively, Mtb was released by macrophages that suffered necrosis and infected and proliferated in other macrophages, thus enabling the spread of the disease, with necrosis being the favorable type of death for this pathogen. Thus, the triggered cell death pathways are likely to be specific to each context along with their consequences, with Mtb as one of the pathogens that can modulate these pathways to their advantage [33].

Microsporidia have been recognized as apoptosis-suppressing pathogens as previously demonstrated for *E. cuniculi* in Vero cells [34], *Anncaliia algerae* in human lung fibroblasts [35], *Nosema bombycis* in ovarian cells from *Bombyx mori* [36], and *Nosema ceranae* in the cells of the ventricular epithelium of *Apis mellifera* [37]. In contrast to these previous findings, the present study demonstrated high percentages of apoptosis occurring in macrophages infected with *E. cuniculi*, especially following pre-incubation with ACs or IACs. We hypothesized that apoptosis may be associated with the release of large amounts of TNF-α, a cytokine that determines apoptosis via the extrinsic pathway that binds to the receptors on macrophages and subsequently activates apoptosis via caspase 8 [38]. In contrast, the accumulated apoptotic cells induce death in the dendritic cells due to pyroptosis or necrosis [39]. Experiments in mice showed that phagocytosis of apoptotic lymphocytes infected with *Trypanosoma cruzi* resulted in a TGF-β- and prostaglandin PGE2-mediated anti-inflammatory response, along with the persistent infection and disease [40]. However, the treatment of mice infected with *T. cruzi* with apoptosis inhibitors reversed the anti-inflammatory profile and reduced the parasite's *ex vivo* replication, corroborating the finding that induction of apoptosis and efferocytosis may be beneficial for microsporidia [41].

Macrophages and other phagocytes are host's first line of defense against microorganisms and are often targets for intracellular pathogens that prevent a microbicidal activity after internalization using antigen-presenting cells as a niche for their survival and replication [42]. Macrophages, with their different profiles, express a defined set of genes that determine different functional patterns. M1 macrophages positively regulate the expression of genes involved in antimicrobial responses and may inhibit efferocytosis partially because of TNF and/or oxidative mediators [43]. M2 macrophages increased their levels of arginase and anti-inflammatory cytokines and reduced the production of pro-inflammatory cytokines and oxygen and nitrogen reactants; however, this profile generally increases in efferocytosis, supporting the elimination of inflammation [44].

In this study, macrophages pre-incubated with ACs showed a higher ratio between CD40$^{high}$/CD206$^{high}$ expression as compared with control, suggesting a greater polarization for the M1 profile. However, infection by *E. cuniculi* directed the macrophages toward the M1 profile with the production of Th1 or pro-inflammatory cytokines. When compared with the control, the expression of CD206 and production of IL-10 increased significantly, revealing that macrophages with an M2 profile were present in the cultures. This finding suggested that the pathogen benefitted from the double macrophage profile for its survival. In addition, the macrophages pre-incubated with IACs showed higher expression of CD206 associated with the release of large amounts of IL-10, indicating an M2 profile that was more permissive for the multiplication of *E. cuniculi*, and acted as a Trojan horse. These ambiguous results, marked by the presence of the M1 and M2 profiles, may reflect *in vivo* conditions as the apoptotic, infected cell efferocytosis determines two simultaneous and conflicting signals, the

presentation of PAMPs causing inflammation and apoptotic cell ligands triggering an anti-inflammatory program [45]. Control macrophages pre-incubated with uninfected (unchallenged) apoptotic cells showed low expression of CD40, CD206, CD80/86, and MHC II compared to macrophages pre-incubated with IACs. Thus, the higher expression of the activation molecules (CD80/86, MHCII) indicated potential microbicidal activity in macrophages pre-incubated with IACS, explaining the presence of less fungal load in these macrophages. One potential mechanism for macrophage activation is through the engagement of costimulatory receptors. These include CD40 and CD80/86 with their respective ligands CD154 and CD28. These interactions are critical for T-cell activation and inflammation in models of adaptive immunity *in vitro* and *in vivo* [46,47], corroborating the results described in this study.

In mice, *Chlamydia pneumoniae* [48] and *Yersinia pestis* [49] are initially phagocytized by neutrophils at the inoculation site, wherein the bacteria either survive or multiply as observed in the case of *Y. pestis*. These apoptotic neutrophils are phagocytosed by macrophages wherein the bacteria replicate. However, they stimulate the release of anti-inflammatory cytokines, which potentially limit the bactericidal activity, corroborating the results observed in our study except for the microbicidal activity not performed in the present study. Alternatively, the macrophages not incubated with ACs showed greater microbicidal activity, characterized by the presence of megasomes, with M1 profile, and demonstrated previously by our group, in the case of infection of peritoneal macrophages by *E. cuniculi* [50]. These findings were similar to those observed in *Helicobacter pylori* infections [51] and *Chlamydia trachomatis* [52].

Efferocytosis of peritoneal or alveolar macrophages suppresses the immune response through the release of anti-inflammatory mediators such as IL-10, NO, TGF-β, PGE2, as well as the inhibition of the synthesis of pro-inflammatory mediators such as TNF-β, GM-CSF, IL-12, IL-1, IL-18, and LTC4 [20,21]. IL-6 is a pleiotropic cytokine that mediates several functions, including the regulation of the immune system by the production of pro- and anti-inflammatory cytokines [53]. Our previous studies demonstrated that under *in vitro* and *in vivo* conditions, the peritoneal macrophages obtained from XID mice and infected in *E. cuniculi* showed a profile of M2 macrophages with predominant death by necrosis, whereas the macrophages of BALB mice revealed a polarization of M1 macrophages and death due to apoptosis and released high levels of IL-6 and IL-10 [50]. This suggested an anti-inflammatory effect, corroborating the pleiotropic behavior of the macrophages. In this study, the production of pro-inflammatory cytokines (TNF and MCP1) associated with the presence of M1 macrophages and the release of anti-inflammatory cytokines (IL-10 and IL-6) with the presence of M2 macrophages reinforced the coexistence and switching of the two macrophage profiles whenever any IAC efferocytosis occurred. The evaluation of these data suggested that the double macrophage profiles occurred both *in vivo* and *in vitro* as also was observed in our *in vitro* study in the group of IACs and high levels of macrophage death by apoptosis, contributing to the multiplication and survival of the pathogen.

These cell death pathways contributed to the defense of the host against microbial infections. Efferocytosis, as a consequence of cell death, could subvert the immune response via the "Trojan horse" mechanism or triggering anti-inflammatory programs that contribute to the evasion of the immune response and persistence of the pathogen. In conclusion, our results showed that efferocytosis of apoptotic cells infected or not had a suppressive effect on the macrophage activity, characterized by the presence of multiplication stages of *E. cuniculi* inside the macrophages. They simultaneously directed the macrophages to an M2 profile with the production of large quantities of IL-10 and IL-6. These data suggested that *E. cuniculi* can exploit efferocytosis to enter the host cell, multiply, and spread, as well as modulate to a less inflamed environment, thereby constituting a mechanism for evading immunity.

## Supporting information

**S1 Fig. Analysis of phagocytosis of apoptotic cells (ACs) or infected apoptotic cells (IACs) marked with CFSE by macrophages.** (A) Percentage of phagocytosis of ACs and IACs. (B) Median CFSE fluorescence in macrophages. T-test tested $^*p < 0.05$ with significance between the groups.
(TIF)

**S2 Fig. Ultramicrography images of macrophages infected with *E. cuniculi*.** (A) Macrophages controls. (B) Macrophages challenged with *E. cuniculi* spores show spores internalized in phagosomes in the cytoplasm (arrow). (C) Macrophages with *E. cuniculi* spores internalized in phagosomes in the cytoplasm (arrow) and with megasomes (Me).
(TIF)

**S3 Fig. Ultramicrography image of macrophages pre-incubated with apoptotic cells.** (A) Macrophages around an apoptotic cell (Ab). (B) Apoptotic bodies (arrows) phagocytized by macrophages. (C) A macrophage emitting pseudopods involving necrotic cell content (Ne).
(TIF)

**S4 Fig. Evaluation of apoptosis induced by UV irradiation.** (A) Dot plot showing the percentages obtained from live (7AAD⁻Annexin⁻), necrotic (7AAD⁻Annexin⁻), apoptotic cells (7AAD⁻Annexin⁺), and late apoptosis (7AAD⁺Annexin⁻). (B) Percentages obtained from apoptosis (apoptosis) and late apoptosis (death) using 1 pulse with 100 J.
(TIF)

**S5 Fig.**
(PNG)

## Acknowledgments

We would like to thank the Parasitology Department of the Federal University of São Paulo, Brazil (Unifesp) that allowed the access and use of UV radiations.

## Author Contributions

**Conceptualization:** Anuska Marcelino Alvares Saraiva, Maria Anete Lallo.

**Data curation:** Luciane Costa Dalboni, Anuska Marcelino Alvares Saraiva, Fabiana Toshie de Camargo Konno, Elizabeth Cristina Perez, Jéssica Feliciana Codeceira, Diva Denelle Spadacci-Morena, Maria Anete Lallo.

**Formal analysis:** Luciane Costa Dalboni, Anuska Marcelino Alvares Saraiva, Elizabeth Cristina Perez, Diva Denelle Spadacci-Morena, Maria Anete Lallo.

**Funding acquisition:** Maria Anete Lallo.

**Investigation:** Anuska Marcelino Alvares Saraiva, Elizabeth Cristina Perez, Diva Denelle Spadacci-Morena, Maria Anete Lallo.

**Methodology:** Luciane Costa Dalboni, Anuska Marcelino Alvares Saraiva, Fabiana Toshie de Camargo Konno, Jéssica Feliciana Codeceira, Diva Denelle Spadacci-Morena, Maria Anete Lallo.

**Project administration:** Maria Anete Lallo.

**Resources:** Diva Denelle Spadacci-Morena, Maria Anete Lallo.

**Supervision:** Maria Anete Lallo.

**Validation:** Diva Denelle Spadacci-Morena, Maria Anete Lallo.

**Writing – original draft:** Luciane Costa Dalboni, Anuska Marcelino Alvares Saraiva, Elizabeth Cristina Perez, Diva Denelle Spadacci-Morena, Maria Anete Lallo.

**Writing – review & editing:** Luciane Costa Dalboni, Anuska Marcelino Alvares Saraiva, Elizabeth Cristina Perez, Maria Anete Lallo.

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
