## [Decision Letter · Decision Letter 0]

4 Dec 2020

PONE-D-20-35590

Encephalitozoon cuniculi  takes advantage of efferocytosis to evade the immune response

PLOS ONE

Dear Dr. Lallo,

Thank you for submitting your manuscript to PLOS ONE. After careful consideration, we feel that it has merit but does not fully meet PLOS ONE’s publication criteria as it currently stands. Therefore, we invite you to submit a revised version of the manuscript that addresses the points raised during the review process.

 As mentioned by Reviewer# 1, please make changes on the manuscript to improve the visibility and clarity. Also, make sure to include scale bars in all images.

We look forward to receiving your revised manuscript.

Kind regards,

Selvakumar Subbian, Ph.D.

Academic Editor

PLOS ONE

Journal Requirements:

Reviewers' comments:

Reviewer's Responses to Questions

**Comments to the Author**

1. Is the manuscript technically sound, and do the data support the conclusions?

Reviewer #1: Yes

Reviewer #2: Yes

2. Has the statistical analysis been performed appropriately and rigorously? 

Reviewer #1: Yes

Reviewer #2: Yes

3. Have the authors made all data underlying the findings in their manuscript fully available?

Reviewer #1: Yes

Reviewer #2: Yes

4. Is the manuscript presented in an intelligible fashion and written in standard English?

Reviewer #1: No

Reviewer #2: Yes

5. Review Comments to the Author

Reviewer #1: Overall, an interesting series of experiments and the data is intriguing. The manuscript needs editing for style and clarity.

Comments

1. In Figure 2 B it is shown that ingestion by macrophages of ACs (either infected or uninfected) leads to those cells having more spores at 24 hours following infection by E. cuniculi, and also that if the macrophage ingested an infected AC (IAC) then the amount of spores at 24 hours following infection by E cuniculi is less than that seen in macrophages that ingest an uninfected AC. This is an interesting observation, but after reading the paper there is no clear mechanism identified that leads to this effect. What is different about an IAC that causes the macrophage to have this lower growth rate for E cuniculi following infection.

2. While an 1 or 2 images of this process at the EM level would be useful the extensive EM presented does not add significant data to the paper. I would suggest that most of these EM images could become supplemental figures and that EM within the actual paper be limited to 1 or 2 images demonstrating the present of IACs and that replication is not occurring following ingestion by the macrophages.

3. The resolution of figures 10 and 11 needs to be improved (line drawings should be 600dpi at this size to be able to clearly read these labels)

4. The data on M2 polarization is interesting, but as noted by the authors (line 551-553) the results were ambiguous with respect to the M1 and M2 profiles. The authors to to refine the text to provide a clear explanation of the results and the implications of this work, especially as it relates to Figure 2B.

5. While increased phagocytosis might explain increased infection, this could also be due to other changes in these macrophages rendering them permissive to infection. Was a control done with heat killed spores? This could provide a way to look at phagocytosis without infection (via the polar tube) as the spores that were heat killed would not germinate and otherwise infect the cells. In addition, phagocytosis rates could also be measured by the uptake of latex beads by the macrophages. It would be useful in understanding figure 2B if some measures of phagocytosis efficiency were provided for macrophages treated with AC and IAC compared to controls untreated macrophages.

Reviewer #2: The authors present clear and elegant evidence of how efferocytosis of infected apoptotic cells shows a suppressive effect on the activity of macrophages with an M2 profile and the production of large amounts of IL-10 and IL-6, suggesting that E. cuniculi can take advantage of efferocytosis to enter the cell, multiply and spread throughout the body and modulate an anti-inflammatory environment, thus constituting a mechanism to avoid the immune response.

The manuscript shows originality and is novel in approaching the subject since few reports of how efferocytosis promotes an anti-inflammatory environment in infections with E. cuniculi; the microscopy images add much weight to the research. I also consider that it provides novel information regarding the immune response in this disease; the results adequately and understandably are presented; they are justified and are related to the objectives clearly and objectively.

improve the quality (sharpness) of the figures, mainly figures 7, 10, and 11. When trying to download figure 10, and figure 9 of the document opens, the link or hyperlink of the image is damaged.

Line 70 class instead of "clade"

6. PLOS authors have the option to publish the peer review history of their article (what does this mean?). If published, this will include your full peer review and any attached files.

Reviewer #1: No

Reviewer #2: **Yes: **Uziel Castillo Velazquez

---

## [Author Response · Author response to Decision Letter 0]

11 Jan 2021

Reviewer's Responses to Questions

We are pleased to resubmit the Manuscript " Encephalitozoon cuniculi takes advantage of efferocytosis to evade the immune response" for revision. We thank the reviewers for their critiques, which have enabled us to sharpen the manuscript considerably. We believe that we have addressed all the points raised by the reviewers and trust that the manuscript is now suitable for publication in PlosOne. A new version was attached as the main file and the corrected version is attached as a supplementary file. Changes are bold and highlighted in red color in the text manuscript, as detailed below. 

4. Is the manuscript presented in an intelligible fashion and written in standard English?

Reviewer #1: No

A. We have provided a new correction to make the text clearer.

Reviewer #2: Yes

5. Review Comments to the Author

Reviewer #1: Overall, an interesting series of experiments and the data is intriguing. The manuscript needs editing for style and clarity.

Comments

1. In Figure 2 B it is shown that ingestion by macrophages of ACs (either infected or uninfected) leads to those cells having more spores at 24 hours following infection by E. cuniculi, and also that if the macrophage ingested an infected AC (IAC) then the amount of spores at 24 hours following infection by E cuniculi is less than that seen in macrophages that ingest an uninfected AC. This is an interesting observation, but after reading the paper there is no clear mechanism identified that leads to this effect. What is different about an IAC that causes the macrophage to have this lower growth rate for E cuniculi following infection.

4. The data on M2 polarization is interesting, but as noted by the authors (line 551-553) the results were ambiguous with respect to the M1 and M2 profiles. The authors to to refine the text to provide a clear explanation of the results and the implications of this work, especially as it relates to Figure 2B.

A. Efferocytosis may eliminate the pathogen or may allow the pathogen to infect the efferocyte in a Trojan-horse type maneuver. The efferocyte will initiate anti-inflammatory or proinflammatory signaling depending upon the combined presence of immune-silencing signals and pro-inflammatory pathogen-associated molecular patterns and damage-associated molecular patterns. 

To answer these questions, we went back to evaluate the groups. When observing the control macrophages pre-incubated with uninfected (unchallenged) apoptotic cells, there was a low expression of CD40, CD206, CD80/86 and MHC II compared to macrophages pre-incubated with IACs. Thus, the greater expression of the activation molecules (CD80/86, MHCII) could indicate potential microbicidal activity in this population, which may explain the finding of a smaller number of spores in these macrophages. Costimulatory molecules are one class of receptors which have been implicated as fulfilling this role in the innate immune response. CD80 and CD86 represent one class of costimulatory receptors. We added a paragraph to explain these interactions and possible explanation for the observed phenomenon.

2. While an 1 or 2 images of this process at the EM level would be useful the extensive EM presented does not add significant data to the paper. I would suggest that most of these EM images could become supplemental figures and that EM within the actual paper be limited to 1 or 2 images demonstrating the present of IACs and that replication is not occurring following ingestion by the macrophages.

3. The resolution of figures 10 and 11 needs to be improved (line drawings should be 600dpi at this size to be able to clearly read these labels)

A. We have reduced the number of figures and improved the quality.

5. While increased phagocytosis might explain increased infection, this could also be due to other changes in these macrophages rendering them permissive to infection. Was a control done with heat killed spores? This could provide a way to look at phagocytosis without infection (via the polar tube) as the spores that were heat killed would not germinate and otherwise infect the cells. In addition, phagocytosis rates could also be measured by the uptake of latex beads by the macrophages. It would be useful in understanding figure 2B if some measures of phagocytosis efficiency were provided for macrophages treated with AC and IAC compared to controls untreated macrophages.

A. In these experiments, we maintained as controls of macrophage activity to observe possible factors independent of efferocytosis, the exclusive challenge with E. cuniculi spores in all phases. Tests with dead spores or latex spheres could also be used to control macrophage activity, however, we consider that the effects related to the pathogen reflect natural events more realistically. Unfortunately in this pandemic moment, we could not repeat the experiments. Our group has been conducting in vitro experiments with E. cuniculi and macrophages of various strains, immortalized as the RAW, and primaries, marrow or peritoneum macrophages. In none of the previous experiments have we observed, so evident, the multiplication of spores inside macrophages with efferocytosis. We would like to reinforce that efferocytosis affects phagocytosis and spore load inside macrophages, but the most striking phenomenon is the multiplication of the pathogen inside macrophages with efferocytosis and M2 profile, events described for the first time in the literature for microsporids, identifying possible evasion mechanism of the immune response

Reviewer #2: The authors present clear and elegant evidence of how efferocytosis of infected apoptotic cells shows a suppressive effect on the activity of macrophages with an M2 profile and the production of large amounts of IL-10 and IL-6, suggesting that E. cuniculi can take advantage of efferocytosis to enter the cell, multiply and spread throughout the body and modulate an anti-inflammatory environment, thus constituting a mechanism to avoid the immune response.

The manuscript shows originality and is novel in approaching the subject since few reports of how efferocytosis promotes an anti-inflammatory environment in infections with E. cuniculi; the microscopy images add much weight to the research. I also consider that it provides novel information regarding the immune response in this disease; the results adequately and understandably are presented; they are justified and are related to the objectives clearly and objectively. Improve the quality (sharpness) of the figures, mainly figures 7, 10, and 11. When trying to download figure 10, and figure 9 of the document opens, the link or hyperlink of the image is damaged.

Line 70 class instead of "clade"

A. We answered the request and changed to class. We have reduced the number of figures and improved the quality.

---

## [Editor Report · Decision Letter 1]

11 Feb 2021

Encephalitozoon cuniculi  takes advantage of efferocytosis to evade the immune response

PONE-D-20-35590R1

Dear Dr. Lallo,

We’re pleased to inform you that your manuscript has been judged scientifically suitable for publication and will be formally accepted for publication once it meets all outstanding technical requirements.

Kind regards,

Selvakumar Subbian, Ph.D.

Academic Editor

PLOS ONE
---

## [Editor Report · Acceptance letter]

26 Feb 2021

PONE-D-20-35590R1 

*Encephalitozoon cuniculi* takes advantage of efferocytosis to evade the immune response 

Dear Dr. Lallo:

I'm pleased to inform you that your manuscript has been deemed suitable for publication in PLOS ONE. Congratulations! Your manuscript is now with our production department. 

Kind regards, 

on behalf of

Dr. Selvakumar Subbian 

Academic Editor

PLOS ONE